# Disinfection of Rainwater for Economic Purposes

**Monika Zdeb** <span>●</span> **and Dorota Papciak** *<span>●</span>

Department of Water Purification and Protection, The Faculty of Civil and Environmental Engineering and Architecture, Rzeszow University of Technology, Al. Powstańców Warszawy 12, 35-959 Rzeszów, Poland; mzdeb@prz.edu.pl
* Correspondence: dpapciak@prz.edu.pl

**Abstract:** Rainwater storage systems are one of the elements of the implementation of sustainable water management. The use of rainwater in households or public buildings reduces the consumption of water supply water for purposes that do not require very good quality water. In crisis situations, rainwater could also be a source of water for drinking and hygiene. In order to use rainwater, it must comply with sanitary quality standards. This paper presents the results of research on the disinfection of rainwater and the possibility of its safe use in the economy as an alternative to tap water. The elements of the proposed pretreatment and disinfection system were selected adequately for the quality of the collected rainwater and its intended use. The aim was to obtain water safe for drinking and hygienic purposes. Rainwater was collected from a roof covered with ceramic tiles, and then subjected to prefiltration, ultrafiltration and disinfection with UV rays. Water before and after treatment was characterized on the basis of a number of microbiological parameters (total number of bacteria at 37 °C and 22 °C; number of coliform bacteria, *Escherichia coli*, *Enterococci*, *Pseudomonas aeruginosa*) and the content of nutrients (TOC, ammonium nitrogen, nitrates, nitrites, phosphates). The use of ultraviolet radiation allowed for the complete removal of indicator bacteria and a significant reduction in the total number of bacteria, from nearly 2500 CFU/mL to 25 CFU/mL for bacteria at 22 °C and from 2010 CFU/mL to 18 CFU/mL for bacteria at 37 °C. The effectiveness of rainwater disinfection, its microbiological stability after disinfection and the time after which the bacterial microflora regenerates, as well as the possibility of using rainwater for drinking and hygienic purposes after disinfection, was determined.

**Keywords:** rainwater; microbiological quality; UV disinfection; use of rainwater; microbiological stability

## 1. Introduction

Both the quantity and quality of water resources are important for the health of the population and economic sectors, which makes water a factor determining the standard of living of society.

Over 733 million people still do not have access to sufficient clean water [1]. Moreover, climate change is causing an intensification of extreme events around the world. The World Meteorological Organization (WMO) reports that from 2000 to today, droughts p have increased by 29% and over 2 billion people suffer from water shortages. It is predicted that by 2050, droughts may affect more than three-quarters of the world's population [2]. At the same time, the number of floods has increased over the last two decades. In 2021, 223 large-area floods occurred [3]. The global increase in the human population is causing a constant increase in water demand. Moreover, it is predicted that by 2050, approximately 64% of people will live in cities [4]. All these factors are an impetus to search for alternative methods of water capture and treatment in order to meet humanity's needs for this resource. The modern model of rainwater management in cities assumes that water should be managed at the place of its origin. Rainwater harvesting systems (RWHS) are still used as a primary source of water supply for millions of people in developing countries [5–8].

However, even in developed countries, regulations and laws are increasingly encouraging rainwater harvesting and reuse as a sustainable solution to improve water supply security [9–12]. Recently, drinking water resources have been limited also due to significant anthropogenic pressure related to water use and the discharge of sanitary sewage into surface and groundwater. Sustainable water and sewage management in urban areas is crucial. First of all, it requires refinement and calibration of rainfall and runoff models for urbanized catchments, with particular emphasis on the main components of the discussed models, i.e., the catchment surface runoff model and the hydraulic model of sewage flow in stormwater or combined sewage pipeline systems [13].

The composition of rainwater is varied [14,15]. It may contain microorganisms that are washed out of the atmosphere and from the roof coverings over which rainwater flows. After entering water, some microorganisms from the air have the ability to exist in the aquatic environment and multiply. Examples of such pathogenic bacteria are *Yersinia pestis, Corynebacterium diphtheriae, Mycobacterium tuberculosis* and *Legionella pneumophila* [16,17]. *Escherichia coli*, coliform bacteria and fecal streptococci are also commonly detected [18–20], as well as pathogenic bacteria such as *Campylobacter, Vibrio, Salmonella, Shigella* and *Pseudomonas* [21–23]. Bacteria of the genus *Helicobacteraceae* were also detected [24]. Protozoa that pose a threat to human health and occur in water include *Cryptosporidium* spp., *Entamoeba histolytica, Giardia intestinalis, Acanthamoeba* spp., *Cyclospora cayetanensis* and *Naegleria fowleri* [25,26]. In addition to the abovementioned bacteria and protozoa, rainwater may contain fungi, most often molds belonging to the oomycete type (Oomycota) and the zygomycete class (*Mucor, Rhizopus*), as well as ascomycota (*Ascomycota*)—both molds (e.g., Penicillium and Aspergillus genus) and yeast, as well as mitosporic fungi (*Deuteromycota*) [27].

The sources of chemical and biological pollution of rainwater are atmospheric air, unpaved surfaces (roofs) and paved surfaces of the catchment area (roads, squares, sidewalks) and other catchment surfaces (arable fields, forests, open areas), including storm and combined sewage systems [28–30].

Collected and stored rainwater can only be used to irrigate crops and for cleaning purposes. Rainwater after purification and disinfection processes can be an economically justified alternative to tap water. Depending on the results obtained, the choice of the final path for rainwater management will be determined by economic calculations. For many years, optimal solutions have been sought to enable the collection and retention of water of a quality that allows for its safe and widespread use. One of the problems that pose a threat to health and requires solution is the bacteriological quality of rainwater [30–32].

One of the basic problems in water treatment, including rainwater, is disinfection and removal of microorganisms, especially those that pose a threat to human health.

The research results presented in this article form a clear and direct response to massive market demand as well as to global climate change and the disturbing hydrological situation. In the future, the development of rainwater treatment technology will allow for reducing the consumption of tap water. Rainwater can be a source of emergency water supply in crisis conditions.

The aim of this work is to disinfect rainwater collected from a roof surface and determine its critical parameters in terms of microbiology at the level set for water intended for drinking, contact with food or hygiene activities, as well as to test the microbiological stability of disinfected rainwater (maintenance of the disinfection effect over time). The novelty of this work is the issue of the stability of rainwater after treatment processes. Attention was paid mainly to microbiological stability determined on the basis of the concentrations of biogenic compounds and the time during which bacterial microflora regenerated in rainwater after disinfection was experimentally checked.

## 2. Materials and Methods

### 2.1. Research Area

This research was conducted in the south-eastern region of Poland: Podkarpackie Province in the foothill part of the region, the Carpathian Foothills. It is characterized by a transitional climate with average annual temperatures of +7 °C (18 °C in summer and −3 °C in winter). The average amount of rainfall is 750 to 800 mm. The dominant winds are south-west, west and north-west. The main elements of air and rainwater pollution are activities of the domestic and municipal sector and traffic emissions (Figure 1).

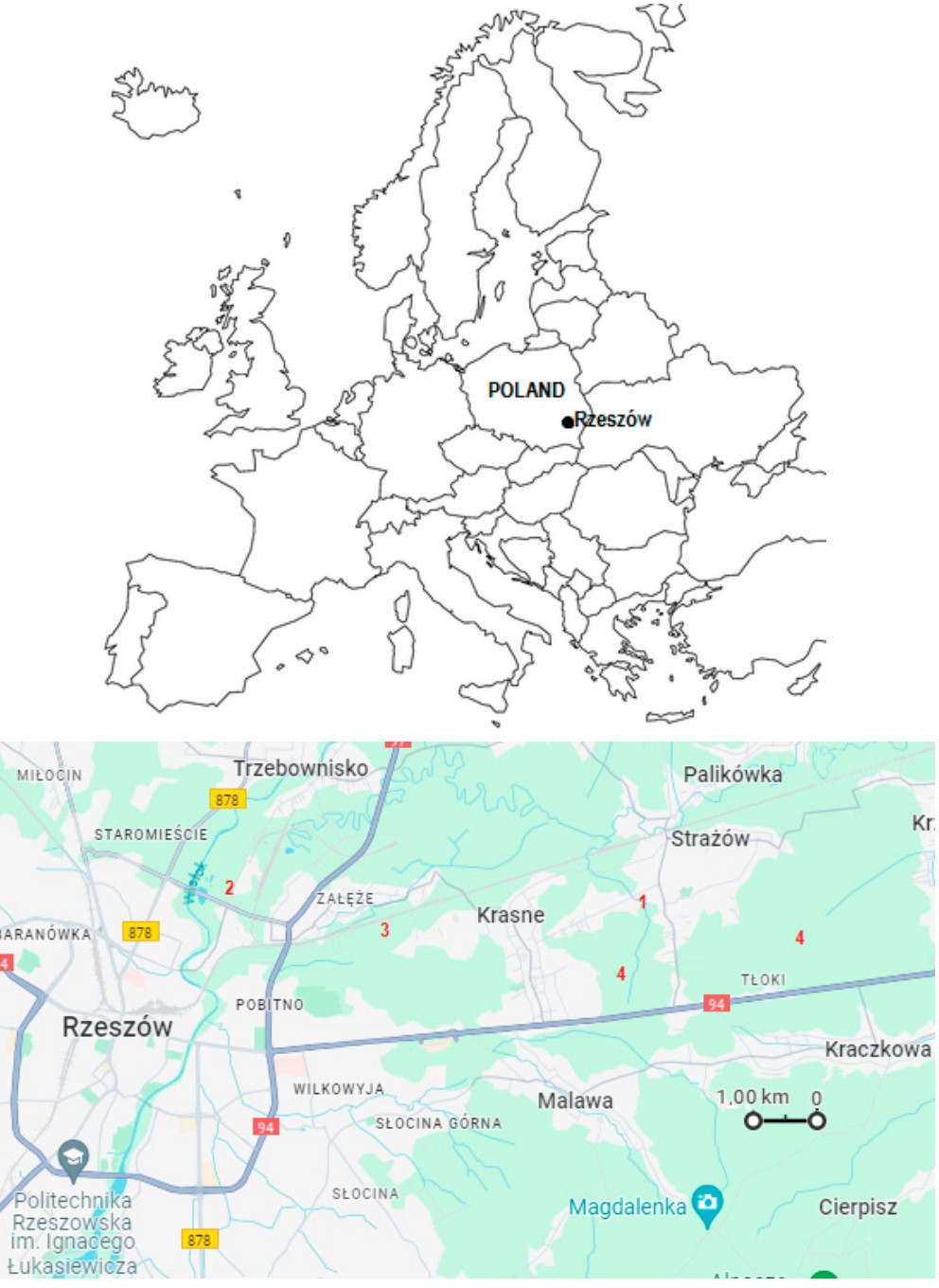

**Figure 1.** Research area (1. rainwater sampling site, 2. municipal sewage treatment plant, 3. municipal waste incineration plant, 4. cultivated fields).

The roof surface of a building located in a suburban housing estate with low-rise single-family buildings was used to collect samples. The roof surface is covered with concrete tiles, with an angle of inclination of 45°. To the west of the sampling site, there are agricultural areas fertilized with minerals (600 m), a sewage treatment plant (6.500 m) and the city of Rzeszow (9.000 m), which may potentially influence the chemical and microbiological composition of rainwater [29,33].

### 2.2. Subject of Research

Rainwater was collected into polyethylene tanks directly from the roof gutter. The collected rainwater was immediately transported to the laboratory and treated in an experimental purification system [34] shown in Figure 2. The effectiveness of removing contaminants causing water turbidity, the effectiveness of eliminating microorganisms by ultrafiltration using a membrane and ceramic filter, and the effectiveness of final disinfection were tested using a UV filament lamp and a UV LED radiator.

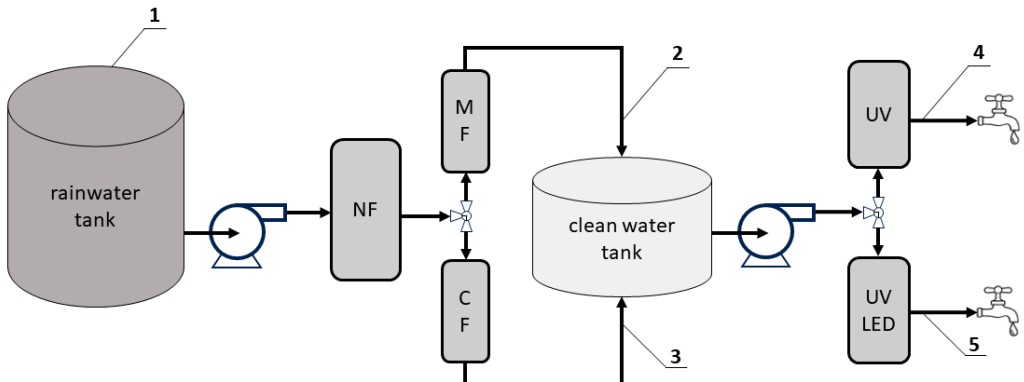

**Figure 2.** Experimental rainwater treatment system: NF—net filter; MF—membrane filter; CF—ceramic filter; UV—UV lamp; UV LED—UV LED radiator; 1 to 5—treated water sampling points.

Rainwater was stored in a preliminary tank, which retained larger suspensions through sedimentation and allowed for maintaining a constant water flow in the treatment system. Rainwater was subjected to preliminary filtration, primary filtration and disinfection.

Prefiltration: This protected the technological system devices. A mesh filter with a filtration threshold of 20 μm was used, which allowed obtaining water with an assumed turbidity value of 1 NTU, and its additional advantage was the possibility of mechanical rinsing and repeated use.

Basic filtration (ultrafiltration): Due to availability, the possibility of scaling and the high efficiency of removal of micropollutants and microorganisms, it was carried out in two variants, using either a membrane filter with a filtration threshold of 0.1 μm or a ceramic filter with a filtration threshold of 0.3 μm.

Final disinfection: This stage had a cleaning function, enabling the production of water of very high bacteriological quality. Disinfection was carried out in two variants, using either a filament UV mercury lamp or a UV LED radiator (Philips production line at the HELIO Bulb Factory, 40-096 Katowice, Poland). The experiment used a Nordic-Tech UV-C bactericidal lamp for water with reaction chamber dimensions of 260 mm × ⌀51 mm and power of 11 W with a PHILIPS filament. The emitted wavelength is 265 nm. For this lamp model, with a nominal flow of 0.20 m³/h, the UVC dose is 300 J/m². The selected model is dedicated to the sterilization of drinking water in small households, as well as in aquariums and small ponds from 3 to 9 m³.

The second source of UV-C radiation was the PearlAqua Micro LED lamp, 12 V. The producer, AquiSense Technologies, (4400 Olympic Blvd, Erlanger, Kentucky 41018 USA) does not provide specific operating parameters or explain how water flows through the system containing the UV-C radiation source. It is a very compact device, intended for specific low-flow applications, e.g., for taps for three-phase filters (reverse osmosis), as

a complement to a reverse osmosis filtration station. A treated water tank was placed between the main filtration and final disinfection components, which ensured coverage of daily water consumption variability. The proposed technological system allows for an efficiency of 20 L/h, ensures health safety of water and can be scaled and modified depending on the requirements and its application.

### 2.3. Research Procedures

The quality of rainwater and treated water in the experimental technological system was assessed based on the parameters and methods included in Tables 1 and 2.

**Table 1.** Methods of assessing the microbiological quality of rainwater.

| Parameter | Unit | Analytical Method/Standard |
| --- | --- | --- |
| The total number of bacteria at 37 °C (*mesophilic bacteria*) | CFU/ml | HTP method using R2A Agar [35] |
| The total number of bacteria at 22 °C (*psychrophilic bacteria*) | CFU/ml | HTP method using R2A Agar [35] |
| **Number of***Escherichia coli* | CFU/100 mL | Membrane filtration procedure using Chromocult® Coliform Agar [36] |
| **Number of***Enterococcus* | CFU/100 mL | Membrane filtration procedure using Slanetz and Bartley Agar [37] |

**Table 2.** Methods of assessing the physicochemical quality of rainwater.

| Parameter | Unit | Analytical Method/Standard |
| --- | --- | --- |
| pH | - | Potentiometric measurement [38] |
| Turbidity | NTU | nephelometric method 2100P ISO TURBIDIMETER HACH [39] |
| TOC | mg C/L | TOC analyzer Sievers 5310 C (SUEZ, Boulder, CO, USA) |
| Ammonium nitrogen | mg $N\text{-}NH_4^+$/L | Spectrophotometric method 8155 (sachet tests—Ammonia Salicylate (1) and Cyanurate (2)) |
| Nitrite nitrogen | mg $N\text{-}NO_2^-$/L | Colorimetric method by Nitrite Test Merck 1.14408 |
| Nitrate nitrogen | mg $N\text{-}NO_3^-$/L | Spectrophotometric method 8039 (sachet tests—NitraVer5) |
| Phosphates | mg $PO_4^{3-}$/L | Spectrophotometric method 8048 (sachet tests—PhosVer3) |

### 2.4. Rainwater Quality and Stability Testing

Water for testing was collected during a period of intense rainfall using the rejection of the first runoff from March to May in 2022–2023 (20 water sample collections). The quality of rainwater was tested before the treatment process (Figure 2, point 1), after the main filtration process (Figure 2, point 2 or 3) and after the disinfection process (Figure 2, point 4 or 5).

The microbiological stability of treated rainwater was tested for water disinfected with a UV filament lamp, stored in polyethylene tanks at 12 °C and 20 °C. The purified water storage temperature was selected to reflect the actual conditions prevailing in the temperate zone of Central Europe in underground or aboveground reservoirs. The stability of treated water was assessed based on microbiological analyses (Table 1), chemical parameters (Table 2) and the stability conditions included in Table 3.

**Table 3.** Water biological stability—recommended values and parameters [40].

| Parameter | Unit | The Condition of Stability |
|---|---|---|
| BDOC * | mg C/L | $\leq 0.25$ |
| $\sum N_{inorg}$ | mg N/L | $\leq 0.2$ |
| $PO_4^{3-}$ | mg $PO_4^{3-}$/L | $\leq 0.03$ |

* BDOC = 7% TOC (Lipphaus i inni 2014).

## 3. Results

### 3.1. Quality of Rainwater Collected from the Roof Surface

The quality of rainwater collected from the surface of a roof covered with concrete tiles is presented in Table 4.

**Table 4.** The quality of tested rainwater collected from roof surface in relation to applicable standards for drinking water. [1] DWS—Polish drinking water standards; [2] DWD—EU Drinking Water Directive; [3] WHO—World Health Organization guidelines for drinking water quality.

| Parameter | Min. | Max. | Mean | Median | DWS [1] | DWD [2] | WHO [3] |
|---|---|---|---|---|---|---|---|
| pH | 7.25 | 7.99 | 7.7 | 7.75 | 6.5–9.5 | 6.5–9.5 | 6.5–8.5 |
| turbidity (NTU) | 1.6 | 5.5 | 2.8 | 2.18 | ac < 1.0 | ac * | 1.0 |
| ammonium nitrogen (mg/dm$^3$) | 0.26 | 0.82 | 0.54 | 0.5 | 0.50 | 0.50 | 0.50 |
| nitrite nitrogen (mg/dm$^3$) | 0.009 | 0.03 | 0.02 | 0.02 | 0.5 | 0.5 | 0.5 |
| nitrate nitrogen (mg/dm$^3$) | 0.2 | 1.2 | 0.65 | 0.5 | 50.0 | 50.0 | 50.0 |
| phosphates (mg/dm$^3$) | 0.01 | 1.48 | 0.15 | 0.04 | - | - | - |
| TOC (mg/dm$^3$) | 1.24 | 3.32 | 2.36 | 2.31 | wac ** | wac | |
| total number of bacteria at 20 °C (CFU/mL) | 390 | 12,200 | 2469 | 1255 | 100 | wac | - |
| total number of bacteria at 37 °C (CFU/mL) | 90 | 16.000 | 2010 | 910 | 20 | - | - |
| the number of *Escherichia coli* bacteria (CFU/100 mL) | 0 | 91 | 18 | 7 | 0 | 0 | 0 |
| the number of *Enterococci* (CFU/100 mL) | 0 | 98 | 34 | 11 | 0 | 0 | 0 |

* acceptable, ** without abnormal changes.

The tested rainwater was characterized by a pH ranging from 7.25 to 7.99 and a turbidity well above 1NTU. It also contained nitrogen, phosphorus and organic carbon compounds. A large total number of bacteria (up to several thousand CFU/mL) was detected in the tested rainwater, as well as the presence of indicator bacteria of fecal contamination—*Escherichia coli*, coliform bacteria and *Enterococci*.

### 3.2. Quality of Rainwater after Filtration and Ultrafiltration

The use of an ultrafiltration membrane significantly reduced the turbidity of rainwater and reduced the total number of bacteria. However, the turbidity in the water after filtration using a ceramic filter increased, and the effectiveness in removing bacteria was lower than when ultrafiltration was used. Despite the assurances of the manufacturers of the selected filtration membranes, in both test variants, single colonies of indicator bacteria of fecal contamination were detected: coliform bacteria and *Escherichia coli* (Table 5).

**Table 5.** The quality of tested rainwater after the ultrafiltration process in relation to applicable standards for drinking water.

| Parameter | Unit | Ultrafiltration Membrane | | | | Ceramic Membrane | | | |
|---|---|---|---|---|---|---|---|---|---|
| | | Min. | Max. | Mean | Median | Min. | Max. | Mean | Median |
| pH | - | 7.12 | 8.21 | 7.6 | 7.66 | - | - | - | - |
| turbidity | NTU | 0.11 | 0.69 | 0.4 | 0.41 | - | - | - | - |
| ammonium nitrogen | mg/dm$^3$ | 0.06 | 1.08 | 0.49 | 0.37 | - | - | - | - |
| nitrite nitrogen | mg/dm$^3$ | 0.006 | 0.03 | 0.01 | 0.02 | - | - | - | - |
| nitrate nitrogen | mg/dm$^3$ | 0.4 | 1.9 | 1.02 | 1 | - | - | - | - |
| phosphates | mg/dm$^3$ | 0.01 | 1.16 | 0.17 | 0.03 | - | - | - | - |
| OWO | mg/dm$^3$ | 1.2 | 4.95 | 2.7 | 2.8 | - | - | - | - |
| total number of bacteria at 20 °C | CFU/ml | 4 | 650 | 213 | 159 | 0 | 607 | 79 | 4 |
| total number of bacteria at 37 °C | CFU/ml | 1 | 220 | 56 | 21 | 0 | 610 | 104 | 4 |
| the number of *Escherichia coli* bacteria | CFU/100 mL | 0 | 1 | 0 | 0 | 0 | 1 | 0 | 0 |
| the number of *Enterococci* | CFU/100 mL | 0 | 0 | 0 | 0 | 0 | 0 | 0 | 0 |

### 3.3. The Quality of Rainwater after the UV Disinfection Process

The tested rainwater collected from the roof surface was subjected to physical disinfection using ultraviolet radiation, preceded by filtration by a mesh filter and an ultrafiltration membrane. The effects of the UV disinfection process depend on the radiation source (Figure 3). UV disinfection using a filament lamp is much more effective than the variant using a UV LED lamp. This is also confirmed by the results of tests conducted for the presence of indicator bacteria (Table 6).

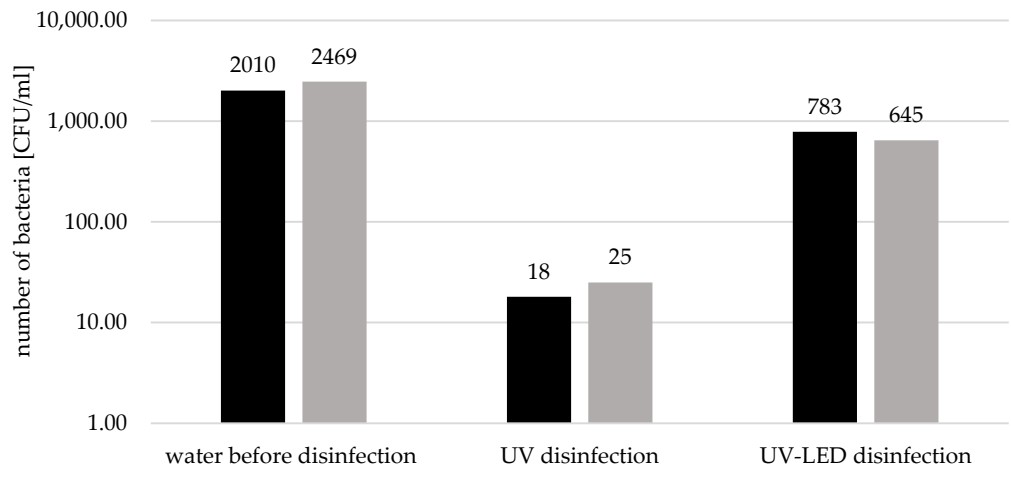

**Figure 3.** Average values of the total number of bacteria in the tested rainwater after UV disinfection.

**Table 6.** Average values of the number of indicator bacteria in disinfected rainwater.

| Parameter | Unit | Number of Bacteria | | |
|---|---|---|---|---|
| | | Rainwater before Disinfection | Rainwater Disinfection by | |
| | | | UV | UV LED |
| Number of coliforms | CFU/100 mL | 10 | 0 | 1 |
| Number of *Escherichia coli* | CFU/100 mL | 18 | 0 | 5 |
| Number of *Enterococci* | CFU/100 mL | 34 | 0 | 1 |

The total number of bacteria at 37 and 20 °C after both the use of a UV LED lamp and a mercury UV lamp was reduced from over 2000 CFU/mL to approximately 700 and 20 CFU/mL, respectively. After using a UV filament lamp, *Escherichia coli* and *Enterococci* were not detected.

### 3.4. Stability of Rainwater after the UV Disinfection Process

The research results clearly indicate that the microflora of disinfected rainwater renews. On the first day of storing water at 12 °C, an increase in the number of bacteria was observed at 37 °C, and in the case of water stored at 20 °C, on the second day (Figures 4 and 5). For the first three weeks, the number of bacteria remains at the level of a dozen or so CFU/mL in a tank kept at 12 °C, and only on the 28th day does it dynamically increase to over 1000 CFU/mL.

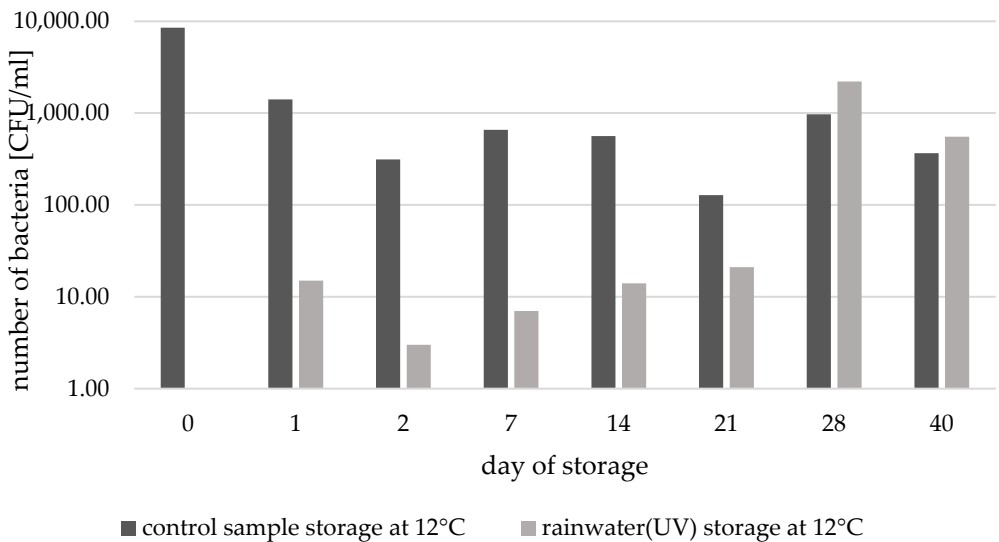

**Figure 4.** Changes in the number of bacteria at 37 °C in rainwater stored at 12 °C after the UV radiation disinfection process.

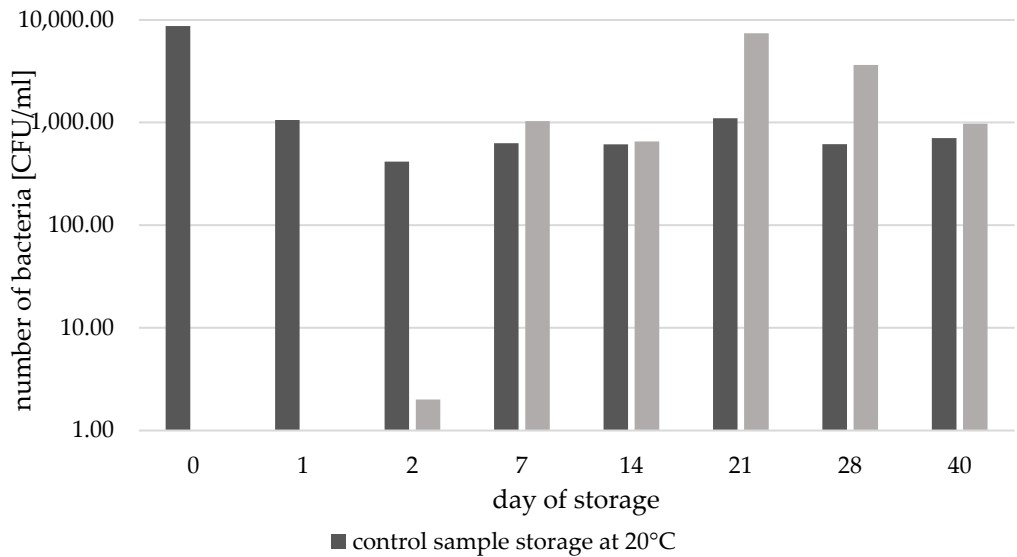

**Figure 5.** Changes in the number of bacteria at 37 °C in rainwater stored at 20 °C after the UV radiation disinfection process.

In a tank maintained at a temperature of 20 °C, rapid multiplication of the recreated bacterial microflora was recorded on the 7th day after UV disinfection. Bacteria at 22 °C

renew 1–2 days after UV disinfection. On the 14th day of storage at 20 °C, their number reached over a thousand CFU/mL (Figures 6 and 7).

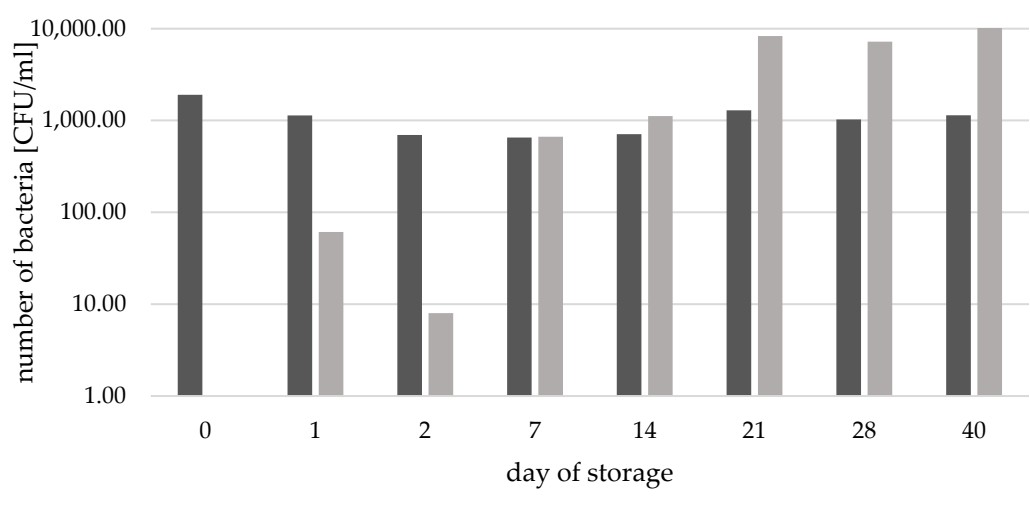

**Figure 6.** Changes in the number of bacteria at 22 °C in rainwater stored at 12 °C after the UV radiation disinfection process.

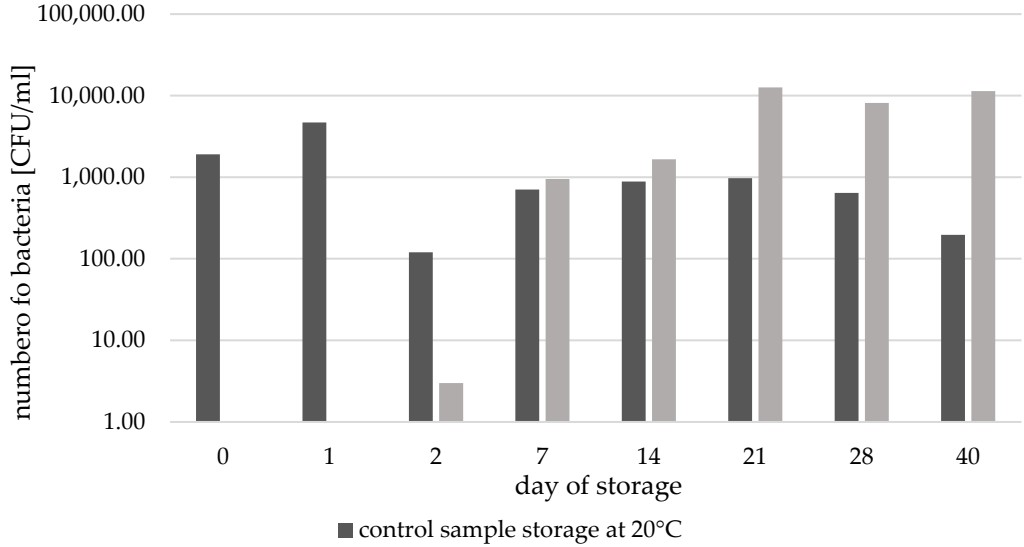

**Figure 7.** Changes in the number of bacteria at 22 °C in rainwater stored at 20 °C after the UV radiation disinfection process.

The population of coliform bacteria and *Escherichia coli* was also renewed, regardless of the rainwater storage temperature after the UV disinfection process (Table 7).

**Table 7.** Changes in the number of indicator bacteria in stored rainwater after the UV disinfection process.

| Day of Storage (after UV Disinfection) | Number of Bacteria (CFU/100 mL) | | | |
|---|---|---|---|---|
| | Coliforms | | *Escherichia coli* | |
| | 12 °C Storage | 20 °C Storage | 12 °C Storage | 20 °C Storage |
| 0 | 0 | 0 | 0 | 0 |
| 1 | 0 | 0 | 0 | 0 |
| 2 | 1 | 0 | 1 | 1 |
| 7 | 0 | 1 | 1 | 2 |
| 14 | 0 | 1 | 0 | 1 |
| 21 | 1 | 1 | 0 | 1 |
| 28 | 0 | 1 | 0 | 0 |
| 40 | 1 | 4 | 0 | 0 |

Parameter values describing biological stability were also determined for the tested rainwater. The quantity of some biogenic substances—phosphorus and nitrogen compounds—exceeded the condition of stability (Table 8).

**Table 8.** Biological stability indicators for tested rainwater.

| Parameter | Unit | The Condition of Stability | Stability Parameters for the Tested Rainwater |
|---|---|---|---|
| BDOC | mg C/L | $\leq 0.25$ | 0.16 |
| $\sum N_{inorg}$ | mg N/L | $\leq 0.2$ | 0.25 |
| $PO_4^{3-}$ | mg $PO_4^{3-}$/L | $\leq 0.03$ | 0.15 |

## 4. Discussion

For many years, research has been conducted on the possibility of replacing tap water with water from alternative sources—primarily using rainwater. It is estimated that rainwater can replace up to 80–90% of tap water consumption. This demand, of course, depends on the building and its purpose (residential, commercial, recreational), as well as the number of people using the water [41,42]. Average annual volumes of water used for plant irrigation depend on the type of vegetation, climate and vegetation period, and may range from 150 to 400 L/m$^2$ [43,44]. Twenty percent of rainwater collected by RWH can be used for washing. Other possible applications in buildings include washing cars and surfaces, powering boilers, extinguishing fires, recovering thermal energy and cooling buildings [45,46]. The multiple demand for water ensures relatively continuous use of rainwater, avoiding its stagnation and problems related to changes in its quality, especially microbiological quality [47,48].

The test results of rainwater collected from the roof surface, presented in the results part of this work, are characterized by physicochemical parameters suitable for drinking water, with the exception of turbidity and ammonium nitrogen concentration. However, the microbiological quality assessment indicates that the rainwater cannot be intended for consumption. This is evidenced by the exceeded permissible number of bacteria at 20 °C and 37 °C and the presence of indicator bacteria of fecal contamination. According to European and national (e.g., Polish) guidelines, the presence of coliform bacteria, *Escherichia coli* bacteria or fecal streptococci is unacceptable in water intended for drinking, contact with food (without thermal treatment) and for hygienic purposes [49,50]. Rainwater therefore requires filtration (to reduce turbidity) and disinfection (removal of microorganisms from the water).

The most frequently used methods to remove or inactivate microorganisms in rainwater are membrane filtration, UV disinfection and chlorination [51,52]. Of course, to achieve maximum efficiency of the process, the method must be appropriately selected for the

quality of the intake water and to ensure ease of use and profitability, which is particularly important when used in households. Both filtration and the use of chlorine-based preparations require experience in operation as well as observation and control of the process—replacing membranes, measuring the demand for chlorine and determining the dose of the preparation. Moreover, chlorination leads to the formation of trihalomethanes and is ineffective against microorganisms associated with suspension, and some microorganisms, such as *Cryptosporidium*, are resistant to its action. The simplest solution in this respect seems to be the use of ultraviolet radiation, most often emitted by mercury filament heaters [53,54].

Despite the assurances of many manufacturers, after the filtration stage (using both the ceramic filter and the ultrafiltration membrane), a significant number of bacteria, including indicator bacteria, were still detected. Ultrafiltration and microfiltration can support and improve the water disinfection process using traditional methods (chlorination, UV radiation, ozonation), because the membrane is a barrier to bacteria, protozoa and even viruses. The size of bacterial cells (0.5–10 μm) and protozoa (3–15 μm) is larger and their complete removal is practically possible using ultrafiltration and microfiltration membranes, because for commercially available membranes, the pore size is usually smaller than 0.3 μm. The size of viruses ranges from 20 to 80 nm, while ultrafiltration membranes have a pore size of about 10–100 nm, so it is theoretically possible to stop them completely. Comparison of the pore sizes of ultrafiltration and microfiltration membranes and the size of microorganisms indicates that the ultrafiltration process theoretically guarantees proper water disinfection. For most microorganisms, removal is estimated at 99.99% reduction. Such high effectiveness was not recorded in the presented study. The overall average bacterial count in the 22 samples decreased from over 1200 CFU/L to just over 200 CFU/L, and the overall bacterial count decreased from 910 CFU/L to 56 CFU/L. This is due to the filter used and its operating time. As the filter operates for longer, its efficiency may decrease. Deposits and biofilm are also formed inside the filter, which may affect the structure of the filter's pores and, as a result, reduce the effectiveness of its operation [55]. Therefore, the processes of removing and deactivating microorganisms in water cannot be based on ultrafiltration alone. Most often, ultraviolet radiation is used as the last stage in water treatment and disinfection systems (including rainwater). Its proven effectiveness allows you to obtain sanitary-safe water. However, excluding filtration and ultrafiltration is unacceptable. A mechanical filter always provides preliminary preparation of the tested rainwater to alter the appropriate elements, e.g., for pH correction or removal of microorganisms. It is also a protective element for the filter of ultrafiltration membranes, by reducing the amount of larger particles and suspensions reaching these elements. Moreover, removing turbidity during filtration is necessary for the effectiveness of the biocidal effect of UV radiators [56,57].

The research conducted in this study shows that UV disinfection using an incandescent lamp is much more effective than the variant using a UV LED lamp. Significant differences in the effectiveness of operation expressed in the total number of bacteria at 22 °C and 37 °C are seen in the power of the compared devices and their sizes, which determine the contact time of water with UV radiation at the same flow rate. The UV LED system used in this research is dedicated to disinfecting small volumes of water intended for drinking.

Only UV disinfection allows obtaining microbiologically safe water, in accordance with national, European and global guidelines on the quality of drinking water [9,10,49,50]. Therefore, it can be used not only for cleaning work or for flushing toilets but also for purposes requiring the highest quality water—for drinking, preparing meals or hygiene activities.

The tested waters also contain biogenic substances, which are one of the basic factors for the multiplication of microorganisms, including pathogenic bacteria, and are one of the basic factors for the secondary multiplication of microorganisms in water after disinfection processes.

The issue of rainwater stability is increasingly discussed and researched. Attempts are being made to determine whether storing rainwater can lead to the elimination of bacteria, especially pathogenic ones. They clearly indicate significant contamination of rainwater stored for even several weeks (without the inflow of fresh portions of rainfall) [45,48].

Stability studies are most often conducted in the context of secondary contamination of tap water in distribution systems [58,59]. The main factors responsible for the secondary growth of microorganisms in water after disinfection processes are biogenic substances and water temperature, the internal condition of the water distribution system network and hydraulic conditions, as well as the presence of disinfectant residues [60,61]. Therefore, obtaining biologically stable water requires not only the production of clean and safe water but also limiting changes during its delivery to the consumer. This is necessary to control the number of microorganisms (including pathogenic ones that pose a threat to human health). In the case of treated and disinfected rainwater, this problem also exists, because such water is most often temporarily stored in tanks, right at the entrance to the installation inside the building, until it is used [16,61]. The results of the research conducted in this study clearly indicate that rainwater is not microbiologically stable, mainly due to the presence of biogenic compounds (Table 5). Nitrogen, phosphorus and carbon compounds, apart from trace elements and water, contribute to the growth and multiplication of bacterial cells [62]. Effective UV disinfection does not protect against the reproduction of bacterial microflora—including indicator bacteria of fecal contamination. This is because exposure to UV radiation can damage the nucleic acids of the cell and reduce enzymatic activity. Over time, microorganisms have managed to develop mechanisms to repair DNA damage [53]. Zhang et al. [62] found that in strains of *Escherichia coli* and Pseudomonas aeruginosa, after exposure to UV radiation, cell membranes remained intact even at UV doses of 300 mJ/cm$^2$. Moreover, dead cells are a source of additional portions of nutrients for living cells [63,64].

## 5. Conclusions

The primary objective of research on the quality and treatment of rainwater is sustainable water management. Appropriate adjustment of rainwater treatment and disinfection methods are the basis for its use as a water source not only for so-called cleaning purposes or watering crops but also for drinking, for hygienic activities and for contact with food. This will allow for a significant spread of the use of rainwater not only in agriculture but also in public buildings and households. The use of ultraviolet disinfection, preceded by ultrafiltration, seems to be the most trouble-free solution and is easy to use, especially for households. This work and its research were focused on testing the simplest to use, cheap, and at the same time effective solutions. The use of the technological system proposed in this work is dedicated to specific applications. Such features make them more willing to be implemented by users in single-family households or within community housing estates. Due to the content of biogenic compounds in the tested waters (even after ultrafiltration) and the regeneration of bacterial microflora, further research directions should concern the development of treatment technologies aimed at achieving microbiological stability of water. Particular attention should be paid to the presence of biogenic compounds present in the treated water and the temperature at which rainwater is stored after the disinfection process.

**Author Contributions:** Conceptualization, D.P. and M.Z.; methodology, D.P. and M.Z.; software, D.P. and M.Z.; validation, D.P. and M.Z.; formal analysis, D.P. and M.Z.; investigation, D.P. and M.Z.; resources, D.P. and M.Z.; data curation, D.P. and M.Z.; writing—original draft preparation, D.P. and M.Z.; writing—review and editing, D.P. and M.Z.; visualization, D.P. and M.Z.; supervision, D.P. and M.Z.; project administration, D.P.; funding acquisition, D.P. All authors have read and agreed to the published version of the manuscript.

**Funding:** This research was funded by the "RainTech" project (N2_022), as part of the Grant Program for R&D works of scientific units under the project entitled "Podkarpackie Center of Innovation" implemented under the Regional Operational Program of the Podkarpackie Voivodeship for 2014–2020, Priority Axis I: Competitive and innovative economy.

**Data Availability Statement:** Data are contained within the article

**Conflicts of Interest:** The authors declare no conflict of interest. The funders had no role in the design of the study; in the collection, analyses, or interpretation of data; in the writing of the manuscript; or in the decision to publish the results.

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
