# Peer review of "Disinfection of Rainwater for Economic Purposes"

_sustainability, doi:10.3390/su152216121_

Round 1

Reviewer 1 Report

Comments and Suggestions for Authors

Thank you for your contributions to this field. Below are some comments: 

1.For Figure 1, it is suggested that re-selecting a large scale map of Rzeszów with feature types such as houses, gardening, etc. where points of interest such as urbanzone, sewage treatment plant, location of the roof surface etc. are clearly visible.

2.The water used for testing was collected from 2022 to 2023 during heavy rainfall between March and May, with a total of 20 water samples collected. Are the analysed data averaged? Was there any pre-processing of the data and did it consider the influence of the environment (for microorganisms)?

3.Why does the ambient temperature of the tank change in the study. Why was 12 degrees, 20 degrees, 22 degrees and 37 degrees varied in this study?

4.The study aims in disinfecting rainwater is economic, especially for applications in the domestic environment. However, there is no discussion or experimentation of the economic costs throughout the manuscript to convey whether this rainwater purification unit, as given by the authors, is economically viable for domestic use. If the cost is not taken into account, there are more effective ways to purify rainwater by electrochemistry.

Author Response

Thank you very much for your insightful review. all comments have been taken into account. Details are included in the attached file. The text of the manuscript has also been supplemented.

Reviewer 2 Report

Comments and Suggestions for Authors

In the reviewed manuscript, the strategy for rainwater disinfection has been proposed. Therefore, the following methods have been chosen to improve quality of rainwater: pre-filtration, ultrafiltration and disinfection with UV rays. In water samples, both physicochemical and biological indicators were monitored. Moreover, the authors evaluated the possibility of using reclaimed water for drinking and hygienic purposes.

In my opinion, the article is suitable for publication in this journal and raises an urgent issue of reusing the rainwater. This problem is particularly important especially in the context of the threat of drought and decreasing water resources. However, at this stage it needs minor revision.

Please include the following remarks in the draft:

1. In the abstract, please highlight the major achievements obtained in the present study.

2. In the introduction section, please also indicate as another factor that limited drinking water resources are combined with unsustainable water and sewerage management in urbanized areas.

Exemplary reference:

Szeląg, B.; Łagód, G.; Musz-Pomorska, A.; Widomski, M.K.; Stránský, D.; Sokáč, M.; Pokrývková, J.; Babko, R. Development of Rainfall-Runoff Models for Sustainable Stormwater Management in Urbanized Catchments. Water 2022, 14, 1997. https://doi.org/10.3390/w14131997

3. At end of the introduction section, please highlight the novelty of the study (after line 86).

4. Please improve the quality of Fig. 1

5. In the discussion section, please provide the scientific explanation of the obtained results in particular in relation to effectiveness in removing bacteria applying the filtration and ultrafiltration methods (observed growths).

6. In my opinion, the positive features of this technology have not been demonstrated well enough in the conclusion section. Moreover, the future perspective can be also presented in this paragraph.

7. Additionally, many editing errors might be found in the reviewed manuscript e.g. table 4 (different colors, commas instead of dots in numbers), figure 7 (incorrect X axis description), formatting units e.g. 237, lack of explanations of abbreviations e.g. RWH.

Author Response

(The authors gave the same response as above.)

Reviewer 3 Report

Comments and Suggestions for Authors

The title could be improved, for example: Disinfection of rainwater as an alternative to tap water.

In the Abstract section, the conclusions could be expressed quantitatively.

Line 44: What does RWH mean?

Line 121: What does NTU mean?

Justify the choice of the bacteria listed in Table 1

Line 155: What does BRWO mean?

Table 4. What does it mean?: DWS, DWD, EU, Lp, ac, OWO, wac, 

Set up Table 1

Table 1, line 8: is it total number?

Line 316: Zhang et al. What year?

It would be good if in section 5, the conclusions were quantified.

Author Response

(The authors gave the same response as above.)
